# Coordinated Unmanned Aircraft System (UAS) and Ground-Based Weather Measurements to Predict Lagrangian Coherent Structures (LCSs)

**DOI:** 10.3390/s18124448

**Published:** 2018-12-15

**Authors:** Peter J. Nolan, James Pinto, Javier González-Rocha, Anders Jensen, Christina N. Vezzi, Sean C. C. Bailey, Gijs de Boer, Constantin Diehl, Roger Laurence, Craig W. Powers, Hosein Foroutan, Shane D. Ross, David G. Schmale

**Affiliations:** 1Department of Biomedical Engineering and Mechanics, Virginia Tech, Blacksburg, VA 24061, USA; pnolan86@vt.edu (P.J.N.); hosein@vt.edu (H.F.); sdross@vt.edu (S.D.R.); 2National Center for Atmospheric Research, Boulder, CO 80305, USA; pinto@ucar.edu (J.P.); ajensen@ucar.edu (A.J.); 3Department of Aerospace and Ocean Engineering, Virginia Tech, Blacksburg, VA 24061, USA; javig86@vt.edu; 4Department of Mechanical Engineering, University of Kentucky, Lexington, KY 40506, USA; christina.vezzi@uky.edu (C.N.V.); sean.bailey@uky.edu (S.C.C.B.); 5Cooperative Institute for Research in Environmental Sciences, University of Colorado Boulder, Boulder, CO 80132, USA; gijs.deboer@colorado.edu; 6UAS Colorado, PO Box 1824, Monument, CO 80132, USA; cdiehl@uascolorado.com; 7Integrated Remote and In Situ Sensing, University of Colorado, Boulder, CO 80132, USA; roger.laurenceiii@colorado.edu; 8Department of Civil and Environmental Engineering, Virginia Tech, Blacksburg, VA 24061, USA; cwpowers@vt.edu; 9School of Plant and Environmental Sciences, Virginia Tech, Blacksburg, VA 24061, USA

**Keywords:** Unmanned Aircraft System (UAS), Lagrangian Coherent Structure (LCS), Weather Research and Forecasting (WRF)

## Abstract

Concentrations of airborne chemical and biological agents from a hazardous release are not spread uniformly. Instead, there are regions of higher concentration, in part due to local atmospheric flow conditions which can attract agents. We equipped a ground station and two rotary-wing unmanned aircraft systems (UASs) with ultrasonic anemometers. Flights reported here were conducted 10 to 15 m above ground level (AGL) at the Leach Airfield in the San Luis Valley, Colorado as part of the Lower Atmospheric Process Studies at Elevation—a Remotely-Piloted Aircraft Team Experiment (LAPSE-RATE) campaign in 2018. The ultrasonic anemometers were used to collect simultaneous measurements of wind speed, wind direction, and temperature in a fixed triangle pattern; each sensor was located at one apex of a triangle with ∼100 to 200 m on each side, depending on the experiment. A WRF-LES model was used to determine the wind field across the sampling domain. Data from the ground-based sensors and the two UASs were used to detect attracting regions (also known as Lagrangian Coherent Structures, or LCSs), which have the potential to transport high concentrations of agents. This unique framework for detection of high concentration regions is based on estimates of the horizontal wind gradient tensor. To our knowledge, our work represents the first direct measurement of an LCS indicator in the atmosphere using a team of sensors. Our ultimate goal is to use environmental data from swarms of sensors to drive transport models of hazardous agents that can lead to real-time proper decisions regarding rapid emergency responses. The integration of real-time data from unmanned assets, advanced mathematical techniques for transport analysis, and predictive models can help assist in emergency response decisions in the future.

## 1. Introduction

Atmospheric wind velocity measurements are critical to air quality [1], weather forecasting [2], and climate studies [3]. Unmanned aircraft systems (UASs) are an emerging technology for atmospheric wind velocity measurements near the surface of Earth [4] where it is difficult and expensive to operate conventional atmospheric sensors reliably. Small UASs, both fixed- and rotary-wing, are low-cost, mobile, and portable with some trade-offs involving flight characteristics. Fixed-wing UASs can fly for periods of around 45 min continuously, but are limited by their flight envelope to open-space operations for launch, maneuvering, and recovery. Rotary-wing UASs can hover, allowing for operations in complex environments, but have limited battery power and generally have shorter flight periods.

Efforts to measure atmospheric properties with UASs began as early as 1971 with [5] using a small fixed-wing platform to carry sensors for direct measurements of atmospheric properties [6]. Similar studies have since followed suit using different mission-specific aircraft designs as detailed in [6]. More recently, indirect approaches have been developed to infer wind velocity using model-based state estimation algorithms. These methods have implemented, among others, the Extended Kalman Filter [7], Unscented Kalman Filter [8], or Finite Horizon Filter [9] to reconstruct wind velocity estimates from inertial and airspeed aircraft sensor measurements. In general, both direct and indirect approaches have yielded promising results as sensor technology continues to advance.

Direct methods of wind estimation encompass the integration of atmospheric flow sensors directly onto the rotary-wing platform [10,11]. This method has been tested using vane [12], solid-state [13,14], hot-wire [10], and sonic anemometers [10,12] as part of sensor placement studies. Results from experiments have demonstrated sensor location to be critical as the propeller downwash can corrupt measurements of ambient wind velocity. Indirect methods, on the other hand, measure wind velocity employing model-free and model-based algorithms. Model-free algorithms render wind velocity measurements from a static relationship between tilt and air-relative velocity [15,16] or the angular kinematics accessed from on-board inertial measurement unit (IMU) sensors [16,17]. Model-based algorithms, in contrast, use a physics-based model along with aircraft state measurements to reconstruct wind velocity using a state observer [18,19,20].

Analyzing atmospheric flows can be challenging due to their chaotic nature. Lagrangian coherent structures (LCSs) have become an increasingly popular tool for the analysis of atmospheric systems. LCSs provide a way to visualize how particles in a flow will evolve; they constitute the skeleton of the flow pattern, particularly regions which are attracting or repelling of nearby fluid, as in Figure 1. For instance, attracting LCSs can correspond to regions of enhanced concentrations of some atmospheric borne chemical species, such as water vapor, pollutants, or hazardous material.

Previous work [21,22,23,24,25,26,27,28] has shown that LCSs tend to coincide with ridges of the finite-time Lyapunov exponent (FTLE) field, which measures the stretching of an air parcel as it advects under the wind. Recently, new Eulerian methods have been developed to detect regions of high attraction and repulsion in fluid flows without the need for simulating air particle paths. These methods are based on the horizontal wind velocity gradient and can be used to calculate an instantaneous approximation to Lagrangian quantities such as LCSs or the FTLE field [29,30,31]. We will take advantage of these Eulerian methods to look for potential LCSs in the experimental results presented here.

Measurements used in this campaign were obtained between 14–19 July 2018 as part of a community-centric field experiment. This experiment was organized in association with the with the International Society for Atmospheric Research using Remotely-Piloted Aircraft (ISARRA) conference held the week before at the University of Colorado Boulder. This “flight week”, titled Lower Atmospheric Process Studies at Elevation—a Remotely-piloted Aircraft Team Experiment (LAPSE-RATE) took place in the San Luis Valley, Colorado. This activity included participation by a variety of university, government, and industry teams. Over the course of six days, over 100 participants supported the coordinated deployment of 50 different unmanned aircraft to complete 1287 total flights, accumulating 262.4 flight hours. These flights were conducted under both Federal Aviation Administration (FAA) Certificates of Authorization (COAs) and FAA Part 107, with the COAs generally supporting flights up to altitudes of 3000 feet above ground level. In addition to the aerial assets, a variety of ground-based observational assets were deployed. These included the Collaborative Lower Atmospheric Mobile Profiling System (CLAMPS), two Doppler LiDAR systems, numerous radiosondes, and mobile surface instrumentation associated with vehicles and small towers. Over the course of the week, flight operations spanned a large ( 3500 km^2^) area over the northern San Luis Valley. The open space around Leach Airfield supported the simultaneous deployment of several aircraft at a time, with these platforms operating alongside several ground-based measurement systems as well as regular radiosonde launches to provide comparison datasets using well-characterized methods and sensors.

In this manuscript, we describe the use of multiple UASs equipped with ultrasonic anemometers to measure wind and temperature and forecast LCSs. The specific objectives of this work were to: develop and deploy multiple UASs equipped with ultrasonic anemometers to measure wind speed, wind direction, and temperature; compare data from the sonic anemometers onboard the UAS against two different ground-based weather stations and a WRF-LES model; and conduct a series of coordinated UAS flights to detect LCSs based on estimates of the horizontal wind gradient tensor. Our ultimate goal is to use environmental data from UASs to drive atmospheric transport models of hazardous agents that can lead to appropriate decisions regarding rapid emergency responses.

## 2. Materials and Methods

### 2.1. Sensor Package Onboard UAS

Two Inspire 2 quadcopters (DJI, Shenzhen, China) were each equipped with an Atmos 22 ultrasonic anemometer (Meter Environment, Pullman, WA, USA) and a Microlog SDI MP/E datalogger (Environmental Measuring Systems, Czech Republic). The Inspire 2 quadcopters were registered with the FAA; registration numbers FA37XL79KC and FA3KHWTTCY. The sensor package was mounted to the airframe of the UAS using carbon fiber rods and custom 3d-printed pieces (found at https://github.com/SchmaleLab/Schmale-Lab-3D-Printing-Files-Nolan-et-al-Sensors-2018). The Atmos 22 weighed 424 grams and was 10 cm in diameter by 16 cm in height. The anemometer measured horizontal wind speed from 0 to 30 m s^−1^ with a resolution of 0.01 m s^−1^ ± 0.3 m s^−1^ or 3% and wind direction from 0 to 359° with a resolution of 1° ± 5°. The datalogger recorded measurements every 15 s. A structure was developed to mount the anemometer to the DJI Inspire 2 using a 30 cm long by 3.8 cm wide polypropylene tube, 3D printed components and carbon fiber tubes Figure 2. The anemometer was mounted to the top of the polypropylene tube and the data logger was mounted to the tube below the anemometer. Four carbon fiber mounting arms attached the pole vertically to the Inspire 2 using stainless steel bolts: one directly below the vertical tube, one to each left and right arm, and one to the rear of the Inspire 2 main body. The data logger was set to record continuously before each flight.

### 2.2. Permissions for Flight Operations

The six counties of the San Luis Valley (SLV), through the support of UAS Colorado and CU Boulder, established a UAS program in 2014 including extensive FAA Permits to operate UASs under public entity Certificates of Authorization (COAs) in a 5,100,000 square mile area of the valley up to 15,000 ft MSL, which equates to about 7500 ft AGL in the central valley. All flights were deconflicted with Local Crop-Dusting Operations, military low altitude training operations along VR routes, routine daily airline operations in and out of KALS, and Local Flight For Life as well as private aircraft operations in the area. UAS pilots for the missions reported in this manuscript were certified Remote Pilots under Part 107; (Schmale, Certificate Number 4038906; González-Rocha, Certificate Number 4010055; and Estridge, Certificate Number #).

### 2.3. Coordinated Aerial and Ground-Based Measurements

#### 2.3.1. Calibration Flight with Vertical Array of Sensors at 10 m (UAS), 4 m (Ground), and 2 m (Ground)

One UAS flight (UAS_B1, Table 1) was conducted over a 4 m flux tower with sensors at fixed heights of 4 m (Atmos 22 sonic anemometer package) and 2 m (CSAT-3 sonic anemometer from Campbell Scientific) (Figure 3). The accuracy of the CSAT-3 was between ±2 and ±6% with ±0.08 m s^−1^ bias precision. Output from the CSAT-3 was logged at 20 Hz via RS232 using a Kangaroo PC portable computer mounted in a weatherproof enclosure. Additional measurements of temperature and humidity were provided at 2 m via a Campbell Scientific E+E Electronik EE181 digital probe (±0.2 °C, ±2.3%RH). This sensor was supplemented by two Campbell Scientific CS215 digital sensors (±0.4 °C, ±4%RH) located 1.5 m and 0.75 m above ground level. All temperature and humidity sensors were housed in a solar radiation shield and logged every 3 s via a Campbell Scientific CR1000X measurement and control data logger. Additional sensors on the flux tower logged every 3 s, but not used in the present study, included a Setra 278 digital barometer, Kipp and Zonen NR-LITE2 Net Radiometer, two Hukeseflux HFP01 Soil Heat Flux Plates, a Campbell Scientific CS655 water content reflectometer, and Campbell Scientific TCAV averaging soil thermocouple probe. All sensors were factory calibrated within one year of use, although intercomparison measurements in a laboratory environment revealed that the EE181 sensor had a consistent 0.5 °C bias which was removed from the measurements reported here.

#### 2.3.2. Calibration Flights with One Ground-Based Sensor at 15 m (MURC) and One UAS at 15 m

Two UASs flights (UAS_A2 and UAS_B2, Table 1) of about 10 min at 15 m AGL were conducted adjacent to the Mobile UAS Research Collaboratory (MURC) tower (Figure 3). The MURC is equipped with a 15 m extendable mast containing several meteorological sensors including a Gill MetPak Pro Base Station that provided barometric pressure, temperature, and humidity; a Gill 3D sonic anemometer for 3D wind measurements; and an R.M. Young Wind Monitor anemometer which provided a redundant horizontal wind measurement.

#### 2.3.3. Simultaneous Flights with Sensors at 15 m in a Triangle Formation (two UASs, and Two Ground Sensors)

Eight coordinated flights (Table 1) were conducted at 15 m AGL in a fixed triangle pattern (each sensor was located at one apex of a triangle with about 100 to 200 m on each side, depending on the experiment) (Figure 4).

### 2.4. WRF-LES Model

Version 3.9.1.1 of the Weather Research and Forecasting (WRF) model [32,33] was used to downscale mesoscale flows to predict the evolution of winds and turbulence in the boundary layer during ISARRA flight week. The model set up was similar to that described by [34]. We used the nesting configuration to downscale operational forecast from 3 km resolution NOAA/NCEP High Resolution Rapid Refresh (HRRR) to a resolution of 111 m using 45 vertical levels. The vertical levels were spaced to maximize resolution in the lowest 2 km of the atmosphere. The nests were configured using one-way feedback (coarse mesh to fine mesh only). Following [34], a refinement ratio of 10 was used between the WRF LES grid and its parent domain in order to minimize the impact of the ‘terra incognita’ range of grid resolutions for which boundary layer parameterizations were not designed [35]. Boundary layer turbulence in D01 was parameterized using the MYNN2 boundary layer while unresolved turbulence in D02 was computed using a sub-grid scale (SGS) closure that includes a prognostic equation for turbulent kinetic energy following [36]. The land surface type was specified using the 20-category MODIS land use dataset. Model forecast data was output at each grid point every 10 min. A higher output rate (0.66 s) was enabled at select grid points that were coincident with ISARRA profiling sites including Leach Airfield. Data used in this study were obtained from WRF-LES runs initialized at 11:00 UTC using HRRR data to initialize and drive the lateral boundaries of the downscaling system. The HRRR is a rapidly-updating forecast system that uses 3DVAR data assimilation to incorporate a wide range of observations to produce a new 18 h forecast every hour [37]. The outer grid of the downscaling system, with 1 km grid spacing, was run for 6 h to spin up dynamically-balanced forcing which was then used to initialize and force the inner-most WRF-LES grid. Data from the forecasts were interpolated to a set of heights above the ground (including 30, 80 and 150 m AGL) and were also interpolated from the Lambert Conformal computational grid onto a regularly spaced grid using bilinear interpolation.

### 2.5. Lagrangian-Eulerian Analysis

Due to their chaotic nature, time-dependent unsteady fluid flows such as atmospheric flows can be challenging to analyze. As mentioned in the introduction, Lagrangian methods such as LCS and the FTLE field have become popular tools to analyze the transport of particles in such flows. Calculating the FTLE field requires Lagrangian data, i.e., numerically simulating the advected paths of fluid particles. The integration of particle trajectories tends to be computationally expensive and necessitates a greater degree of spatial and temporal information than can reasonably be gathered by operators in the field.

New Eulerian tools have recently been developed which use velocity gradients, instead of integrating particle trajectories. This allows for flows to be analyzed by pointwise measurements at as few as three points. These velocity gradients are assembled into the Eulerian rate-of-strain tensor, S in Equation (Equation 3) discussed below. In [30] it is shown that S can provide an instantaneous approximation of the Lagrangian dynamics of a fluid flow. Ref. [30] further states that we should seek objective Eulerian coherent structures (OECSs) based on the invariants of S, as short-term limits of LCSs. Further work on this topic [31] has also shown that in two-dimensional flows, the eigenvalues of S, s_1_ < s_2_, are the limits of the backward-time and forward-time FTLE fields as integration (advection) time goes to zero. Ref. [31] further posits that troughs of the s_1_ field can be identified as instantaneous attracting LCSs and ridges of the s_2_ field can be identified as instantaneous repelling LCSs. For the remainder of this manuscript we shall refer to s_1_ as the attraction rate and s_2_ as the repulsion rate.

For our analysis we be considering the fluid particle advection dynamical system,
(1)$$\begin{array}{c} \frac{d}{dt}x=v\left(x,t\right), \end{array}$$
(2)$$\begin{array}{c} x_{0}=x\left(t_{0}\right). \end{array}$$


In this system x(t) is the position vector of a fluid parcel at time *t* and v(x,t) is the horizontal wind velocity vector at position x(t), time *t*. We define the components of the horizontal position vector, x=(x,y), where *x* is the eastward position and *y* is the northward position, measured either in meters with respect to some convenient reference point or in longitude and latitude, respectively. We will analyze this system by looking at the attraction rate, which is the minimum eigenvalue, s_1_, of the Eulerian rate-of-strain tensor, S(x,t). The Eulerian rate-of-strain tensor is defined based on the horizontal wind gradient,
∇v(x,t),
as
(3)$$S\left(x,t\right)=\frac{1}{2}\left(\nabla v\left(x,t\right)+\nabla v{\left(x,t\right)}^{T}\right).$$


As stated before, the attraction rate provides a means of identifying the attracting OECSs, which are the instantaneous LCSs. The attraction rate provides information on where material particles will converge (Figure 1). The lower the value of the (negative) attraction rate, the more particles will be attracted to that point. We focus on the attraction rate given its importance for predicting enhanced concentrations of atmospherically advected tracers, as nearby particles will converge onto those features and flow with them as opposed to repelling features which particles will diverge from before flowing independent of those features.

### 2.6. Computation of Wind Gradient and Attraction Rate

To calculate the attraction rate we first needed to calculate the gradient of the wind velocity field, ∇v(x,t), for the spatiotemporally varying wind velocity vector v(x,t)=(u,v), where *u* is the eastward wind component and *v* is the northward wind component. For an estimate of the gradient, three measurements of the wind velocity were simultaneously recorded by two UASs and one ground station. The wind velocity data taken from these measurements was then interpolated to a fourth point between the three sensors, Figure 5.

This point is chosen to be along a north-south line with one of the sensors and an east-west line with another. A depiction of the true situation based on a satellite photo can be seen in Figure 6.

The velocity was interpolated to the fourth point using linear interpolation provided by the griddata routine from Python’s SciPy module. Once the velocity was interpolated, the gradient of the velocity field was calculated using a finite-difference scheme between the velocity at the interpolated point and the velocity from the sensors directly north/south and east/west. For example, with the setup shown in Figure 5, the gradient of *u* is calculated as
(4)$$\frac{\partial u}{\partial x}\approx \frac{u_{interp}-u_{B}}{dx},\;\frac{\partial u}{\partial y}\approx \frac{u_{interp}-u_{C}}{dy},$$
where $u_{interp}$ is *u* at the interpolated point, $u_{B}$ is *u* at sensor B, and $u_{C}$ is *u* at sensor C. This method could then be applied to *v* as well to get the full horizontal gradient of the wind vector,
(5)$$\nabla v=\left[\begin{array}{cc} \frac{\partial u}{\partial x} & \frac{\partial u}{\partial y}\\ \frac{\partial v}{\partial x} & \frac{\partial v}{\partial y} \end{array}\right],$$
and the Eulerian rate-of-strain tensor,
(6)$$S=\left[\begin{array}{cc} \frac{\partial u}{\partial x} & \frac{1}{2}\left(\frac{\partial u}{\partial y}+\frac{\partial v}{\partial x}\right)\\ \frac{1}{2}\left(\frac{\partial u}{\partial y}+\frac{\partial v}{\partial x}\right) & \frac{\partial v}{\partial y} \end{array}\right].$$


The attraction rate, $s_{1}$ is then given analytically by
(7)$$s_{1}=\frac{1}{2}\left(\frac{\partial u}{\partial x}+\frac{\partial v}{\partial y}\right)-\frac{1}{2}\sqrt{{\left(\frac{\partial u}{\partial x}-\frac{\partial v}{\partial y}\right)}^{2}+{\left(\frac{\partial u}{\partial y}+\frac{\partial v}{\partial x}\right)}^{2}}.$$


### 2.7. Uncertainty Analysis

We can quantify the uncertainty in our gradient approximation as follows. For example, for the gradient component, $\frac{\partial u}{\partial x}$, we can estimate the uncertainty $\delta \left(\frac{\partial u}{\partial x}\right)$ using (Equation 4) as,
(8)$$\begin{array}{cc} \delta \left(\frac{\partial u}{\partial x}\right) & =\left|\frac{\partial (\frac{\partial u}{\partial x})}{\partial u_{interp}}\right|\delta \left(u_{interp}\right)+\left|\frac{\partial (\frac{\partial u}{\partial x})}{\partial u_{B}}\right|\delta \left(u_{B}\right)+\left|\frac{\partial (\frac{\partial u}{\partial x})}{\partial (dx)}\right|\delta \left(dx\right)\\ & =\frac{1}{dx}\delta \left(u_{interp}\right)+\frac{1}{dx}\delta \left(u_{B}\right)+\frac{1}{dx^{2}}\left|u_{interp}-u_{B}\right|\delta \left(dx\right)\\ & =\frac{1}{dx}\left(\delta \left(u_{interp}\right)+\delta \left(u_{B}\right)+\left|\frac{\partial u}{\partial x}\right|\delta \left(dx\right)\right). \end{array}$$
where δ(·) denotes the uncertainty in the measured quantity. SciPy’s griddata routine uses a barycentric interpolation scheme for linear interpolation, thus we can rewrite $u_{interp}$ as,
(9)$$u_{interp}=c_{1}u_{A}+c_{2}u_{B}+c_{3}u_{C},$$
subject to the constraint that $c_{1}+c_{2}+c_{3}=1$. So, since the anemometers all have the same error, $\delta \left(u_{A}\right)=\delta \left(u_{B}\right)=\delta \left(u_{C}\right)$, we have
(10)$$\delta \left(u_{interp}\right)=\delta \left(u_{A}\right)$$
and
(11)$$\begin{array}{cc} \delta \left(\frac{\partial u}{\partial x}\right) & =\frac{1}{dx}\left(c_{1}\delta \left(u_{A}\right)+\left(1+c_{2}\right)\delta \left(u_{B}\right)+c_{3}\delta \left(u_{C}\right)+\left|\frac{\partial u}{\partial x}\right|\delta \left(dx\right)\right)\\ & =\frac{1}{dx}\left(2\delta \left(u_{A}\right)+\left|\frac{\partial u}{\partial x}\right|\delta \left(dx\right)\right) \end{array}$$


Similar results hold for the other components of the velocity gradient (Equation 5).

We can also determine the uncertainty in the attraction rate $s_{1}$, based on (Equation 7), as
(12)$$\delta \left(s_{1}\right)=\left|\frac{\partial s_{1}}{\partial (\frac{\partial u}{\partial x})}\right|\delta \left(\frac{\partial u}{\partial x}\right)+\left|\frac{\partial s_{1}}{\partial (\frac{\partial u}{\partial y})}\right|\delta \left(\frac{\partial u}{\partial y}\right)+\left|\frac{\partial s_{1}}{\partial (\frac{\partial v}{\partial x})}\right|\delta \left(\frac{\partial v}{\partial x}\right)+\left|\frac{\partial s_{1}}{\partial (\frac{\partial v}{\partial y})}\right|\delta \left(\frac{\partial v}{\partial y}\right)$$
where
(13)$$\begin{array}{cc} \frac{\partial s_{1}}{\partial \left(\frac{\partial u}{\partial x}\right)} & =\frac{1}{2}-\frac{1}{2c}\left(\frac{\partial u}{\partial x}-\frac{\partial v}{\partial y}\right),\\ \frac{\partial s_{1}}{\partial \left(\frac{\partial v}{\partial y}\right)} & =\frac{1}{2}+\frac{1}{2c}\left(\frac{\partial u}{\partial x}-\frac{\partial v}{\partial y}\right),\\ \frac{\partial s_{1}}{\partial \left(\frac{\partial u}{\partial y}\right)} & =\frac{\partial s_{1}}{\partial \left(\frac{\partial v}{\partial x}\right)}=-\frac{1}{2c}\left(\frac{\partial u}{\partial y}+\frac{\partial v}{\partial x}\right), \end{array}$$
and
(14)$$c=\sqrt{{\left(\frac{\partial u}{\partial x}-\frac{\partial v}{\partial y}\right)}^{2}+{\left(\frac{\partial u}{\partial y}+\frac{\partial v}{\partial x}\right)}^{2}}.$$


## 3. Results

UAS flights were conducted between 10 and 15 m AGL at the Leach Airfield in the San Luis Valley, Colorado as part of the ISARRA 2018 flight campaign (Table 1). The UASs were used to collect simultaneous measurements of wind speed, wind direction, and temperature in a fixed triangle pattern (each sensor was located at one apex of a triangle with 100 to 200 m on each side, depending on the experiment, Figure 5 and Figure 6. In addition, high resolution atmospheric simulations using weather research and forecasting (WRF) model large eddy simulation (LES) was used to determine the 4D (space and time) wind field across the sampling domain. Data from the ground-based sensors and the two UASs were used to detect LCSs.

### 3.1. Comparison of Measurements

#### 3.1.1. Calibration Flights

Calibration flights were conducted to compare wind velocity and temperature measurements from UAS A and B to measurements from independent sensors installed at 2, 4 and 10 m AGL as shown in Figure 3. The wind velocity and temperature independent sensors consisted of a CSAT3 sonic anemometer installed at 2 m, an Atmos22 sonic anemometer placed at 4 m or 15 m (on the MURC’s tower), and the Gill 3D sonic anemometer mounted atop of the MURC’s tower at 15 m. For comparison, measurements of temperature and wind velocity recorded at 15 m AGL were considered. Results from this analysis were used as a confidence benchmark for UAS-based measurements of wind velocity and temperature sensors.

#### 3.1.2. Wind Speed

In this section, we present results from measurements of wind speed and direction collected on 13 July 2018. The wind conditions on this day were variable ranging between 2 and 10 m s^−1^ as shown in Figure 7. Atmospheric sampling involved four coordinated UAS missions, comprising eight distinct flights, along with measurements from the 15m_tower_Atmos22, 15m_tower_MURC_3Dsonic, and the 2m_tower_CSAT3. Figure 7 shows the general trend in wind velocity as recorded by independent sensors at 2, 4 and 10 m above ground level during a 10-min interval. The wind speed trend with height is consistent with a power law with coefficient α≈0.2. In Figure 8, measurements of wind speed and direction from UAS B and MURC were compared at 15 m AGL. Agreement for wind speed and direction were determined using a root-mean-squared (RMS) error metric. Results show an RMS error of 0.75 m s^−1^ and 8.9° for wind speed and direction, respectively.

In Figure 9 and Figure 10 wind speed measurements from multiple sensors are displayed. Figure 9 shows the measurements from the ground-based 15m_tower_MURC_3Dsonic (blue) and 15m_tower_Atmos22 (orange), these are overlaid with measurements from the 15m_UAS_A_Atmos22 (black). Figure 10 shows the measurements from the ground-based 15m_tower_MURC_3Dsonic (blue) and 15m_tower_Atmos22 (orange), these are overlaid with measurements from the 15m_UAS_B_Atmos22 (black). The UAS A flights shown are 22, 23, 25, 26. The UAS B flights shown are 9, 10, 11, 12. Details regarding the flights can be found in Table 1. Pearson correlation coefficients for these wind speed measurements range from 0.868 to 0.970 and can be found in Table 2.

In Figure 11, we show the wind speed measurements associated with our attraction rate calculations (orange) along with wind speed predictions from WRF-LES model (blue), these are overlaid with mission averages for the sensor measurements (black). Mission averages are included for the period over which all three sensors were operating. For this comparison, a temporal resolution of 0.66 s was used for the WRF-LES model output. To calculate the wind speed, measurements were taken from two concurrent UAS flights as well as the 15m_tower_MURC_3Dsonic and interpolated to the point where the attraction rate was computed, Figure 5 and Figure 6. The wind speed from the WRF-LES model came from the grid point nearest to where the attraction rate was calculated.

### 3.2. Attraction Rate Measurements

In this section, we present our results for attraction rate as calculated from our wind velocity measurements using the UAS and ground station method described in Section 2.6. As a comparison we show the attraction rate as calculated from the WRF-LES model predictions. To get a picture of what was happening on a larger scale, we also calculated the attraction rate over the San Luis Valley using the 10 m velocity field from the WRF-LES model.

In Figure 12, we show the attraction rate as calculated from the measurements provided by the two UASs and the 15m_tower_MURC_3Dsonic (orange) along with the attraction rate calculated from WRF-LES model predictions (blue), these are overlaid with mission averages for the sensor measurements (black). The uncertainty ranges for the attraction rate measurements are shown in gray. Mission averages are averages for the period over which all 3 sensors were operating. For this comparison the WRF-LES model data was at a temporal resolution of 0.66 s. The attraction rate from the WRF-LES model’s wind predictions was calculated using a central finite-difference scheme from a five point stencil centered on the grid point nearest to where the attraction rate was calculated from the UASs and the 15m_tower_MURC_3Dsonic measurements. In yellow we highlight the time periods around the predicted attraction rate fields shown in Figure 13 and Figure 14. We display the gradients that were used to calculate attraction rate in Figure 15. The gradients from the WRF-LES model (blue) are overlaid with those from our sensors (orange). We also show the range of uncertainty for our sensor gradient calculations in gray.

In Figure 13 and Figure 14, we show the attraction rate field over the San Luis valley on 17 July 2018 as calculated from the WRF-LES model’s 10 m velocity field prediction. In Figure 13, we show the attraction rate field at 1400 MDT. This time was chosen to display due to a large discrepancy between the WRF-LES model’s attraction rate prediction and the attraction rate as calculated from real-world data. In Figure 14, we show the attraction rate field at 1550MDT. This time was chosen because an attracting front was passing through our sampling region out of the east. After the front passes the field is noticeable smoother. In both figures, the point where the attraction rate was calculated for the time series in Figure 12 is shown as a red dot. An animation of the attracting rate field over the San Luis valley can be found at https://youtu.be/jui5GfehWGg.

## 4. Discussion

Concentrations of airborne chemical and biological agents from a hazardous release are not spread uniformly. Instead, there are regions of higher concentration, in part due to local atmospheric flow conditions which can attract agents [24,27,38,39,40]. New tools and technology are needed to monitor and forecast atmospheric transport phenomena [20]. Here, we have described a series of unique field experiments to collect simultaneous measurements of wind speed, wind direction, and temperature using multiple UASs and ground stations. Data from these sensors were compared to a WRF-LES model, and were used to forecast LCS.

Overall, the WRF-LES model provided fairly accurate predictions of both the winds and the attraction rate with some caveats. The wind speed predictions from the WRF-LES model followed the general trend measured by the sensors with both modeled and observed winds ranging between 0 and 4 m s^−1^ during the early afternoon increasing late in the day in response to the development of moist convection and gusty outflows. Modeling the exact timing of deep moist convection and associate gusty winds at a single grid point is not possible, but rather, can be determined in a statistical sense by compositing forecast information across a much larger area of similar surface type. As seen in the attraction rate field (Figure 13 and Figure 14), visual inspection of the modeled 10 m winds reveals very localized convective circulations that resulted in gusty higher winds at the grid point closest to the measurement site. However, within 5 km the 10 m winds were still below 4 m s^−1^ as observed through 15:45 MDT (Figure 9 and Figure 10). Quantifying this spatial and temporal variability can be captured through ensemble approaches which can be derived using spatial statistics and/or by running a multi-member ensemble; however, such analyses are beyond the scope of this paper.

Analyzing the time series data, we see that the attraction rate has quick repeated dips in it, indicating short bursts of attracting activity. Comparing this to the the attraction rate fields shown in Figure 13, we can see that the convective cells are bordered by narrow troughs of the attraction rate field. These dips appear to be an indicator of the movement of convective cells across a location, and thus a transition of the observer from one convective cell to another.

As mentioned before, there was good agreement between the attraction rate as calculated from the WRF-LES model’s predictions and the attraction rate as calculated from sensor measurements. There is an exception to this agreement during the second UAS mission on 17 July 2018 around 1400 MDT. During this mission, the sensors measured the attraction rate drop below −200 hr^−1^, yet the model prediction for this time was closer to −50 hr^−1^. Looking at the attraction rate field for this time period, Figure 13, we can see there was a lot of convective activity going on in this region of the domain. Furthermore, looking at the time series data, Figure 12, we can see that the WRF-LES model does predict a dip in the attraction rate around that time. These dips appear to be an indication of transition between convective cells. Thus, the UAS and ground station measurements are likely picking up highly localized attraction at that time as a convective cell passes by, attraction which is falling below the scale of the model.

Another noteworthy event happened during the fourth UAS flight on 17 July 2018. During this flight, a front passes through the sampling area, coming out of the east. In Figure 14, we show the predicted front (white curve) just before passing by our sampling area, marked as a red dot. In the time series data, Figure 12, we can see two small dips right before 1600 in the attraction rate as predicted by the WRF-LES model. During this same period, we have a sharp drop in the attraction rate as calculated by our UAS measurements, followed by a quick uptick and then another drop as the flight data ends. These dips happen at approximately the same time the front was predicted to pass through the sampling area. It is thus very likely that the drops calculated correspond to the predicted front passing through our measurement sampling area.

The troughs of the attraction rate field that we are detecting are very likely to be important indicators of LCSs. The attraction rate field is the limit of the backward-time FTLE field as integration time goes to zero [31]. As mentioned in the introduction, ridges of the FTLE field tend to coincide with LCSs [21,22,23,24,25,26,27,28]. Whereas the FTLE is defined as a positively valued scalar field, indicating stretching, the attracting rate field is largely negative, indicating shrinking. Troughs of the attraction rate field are the analogues to ridges of the backward-time FTLE field. These troughs can be thought of as attracting LCSs. Therefore strong dips in the attraction rate time series should correspond to the passage of attracting LCSs.

## 5. Conclusions

We equipped a ground station and two unmanned aircraft systems (UASs) with identical ultrasonic anemometers. Flights reported here were conducted 10 to 15 m above ground level (AGL) at the Leach Airfield in the San Luis Valley, Colorado as part of the ISARRA 2018 flight campaign. The ultrasonic anemometers were used to collect simultaneous measurements of wind speed and wind direction in a fixed triangle pattern. Results showed excellent agreement among sensors across different platforms, particularly for wind speed. Over the same time period as the sampling campaign, a WRF-LES model was used to determine the wind field across the sampling domain.

Data from the ground-based sensors and the two UASs were used to detect attracting regions (also known as Lagrangian coherent structures or LCSs), which have the potential to attract and transport high concentrations of chemical and biological agents. This is the first time that direct measurement of an LCS indicator was made in the atmosphere using a team of sensors.

Coordinated teams of aerial and ground-based sensors provide unique environmental data that have the potential to inform real-time decisions regarding rapid emergency responses, such as following the transport of hazardous agents after a hurricane. The integration of real-time data from unmanned assets, advanced mathematical techniques for transport analysis, and predictive models can help assist in emergency response decisions in the future.

## Figures and Tables

**Figure 1 sensors-18-04448-f001:**
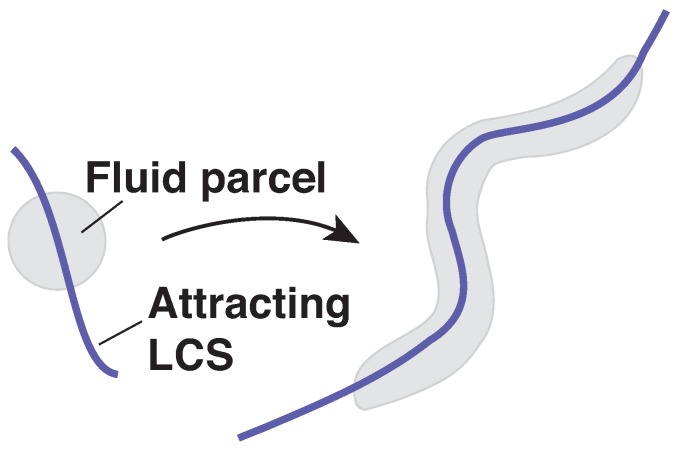
Schematic illustration of an attracting LCS, and its effect on a fluid parcel over a short advection time.

**Figure 2 sensors-18-04448-f002:**
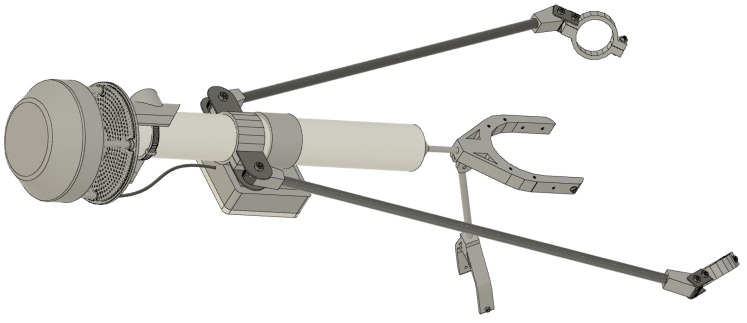
Schematic of Inspire 2 sampler assembly. An interactive 3D version can be found at https://a360.co/2OnKTl4.

**Figure 3 sensors-18-04448-f003:**
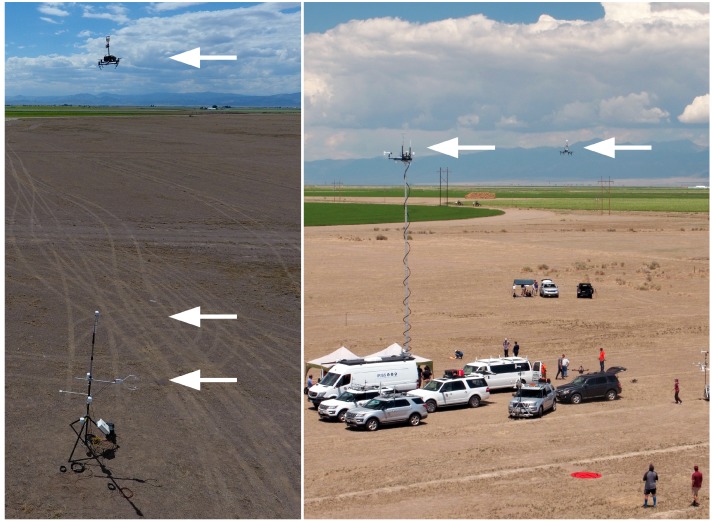
Calibration flight with vertical array of sensors at 10 m (UAS with Atmos 22 sonic anemometer), 4 m (Atmos22 sonic anemometer), and 2 m (CSAT3 sonic anemometer) (**left**). White arrows indicate the sensors at each of the respective heights. Calibration flight (B2) with one ground-based sensor on the MURC tower and one UAS, both at 15 m (**right**). White arrows indicate the sensors for each of the platforms.

**Figure 4 sensors-18-04448-f004:**
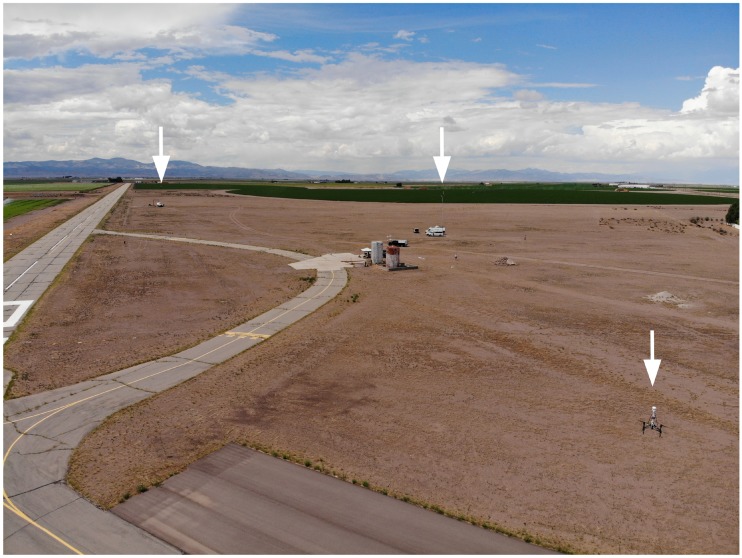
Simultaneous flights with sensors in a triangle formation (white arrows). Both UASs were operating off of marks on the taxiway, and the MURC tower was stationed north of the UAS operations.

**Figure 5 sensors-18-04448-f005:**
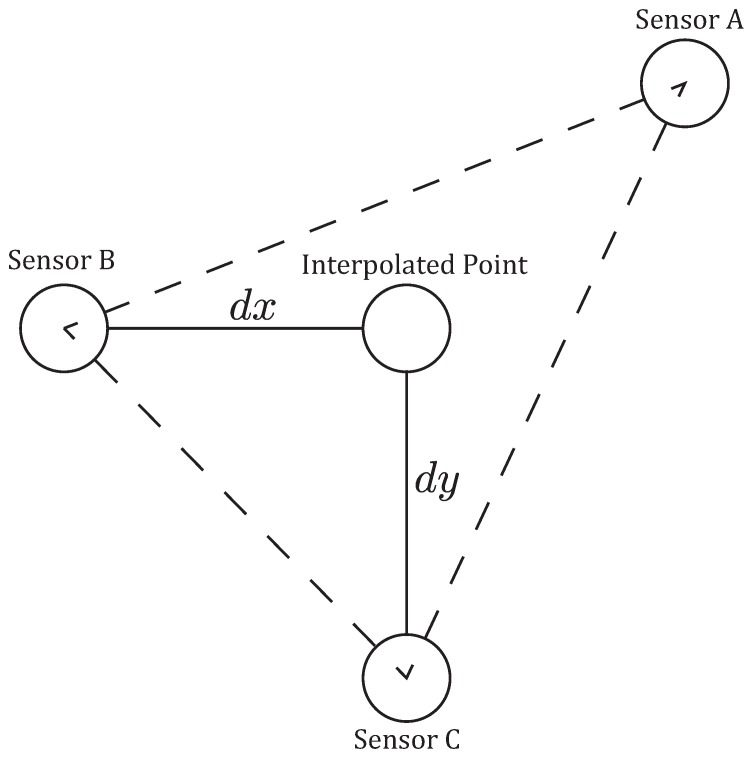
Schematic of how the velocity gradient, ∇v(x,t), was computed from sensor measurements. Using wind measurements from three independent sensors, a linear function was generated for a triangular element. This function was then used to interpolate the wind to an interpolated point. A finite-difference scheme, Equation (Equation 4), was then used to calculate the gradient of the two components, *u* and *v*, of the horizontal wind.

**Figure 6 sensors-18-04448-f006:**
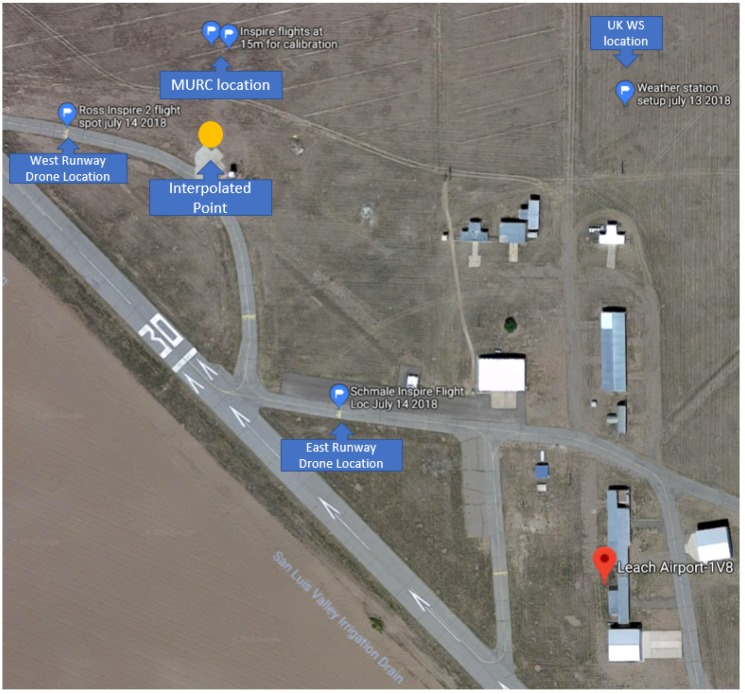
Satellite image of the sampling region with sensor locations marked with blue pins. A potential interpolated point is marked in yellow.

**Figure 7 sensors-18-04448-f007:**
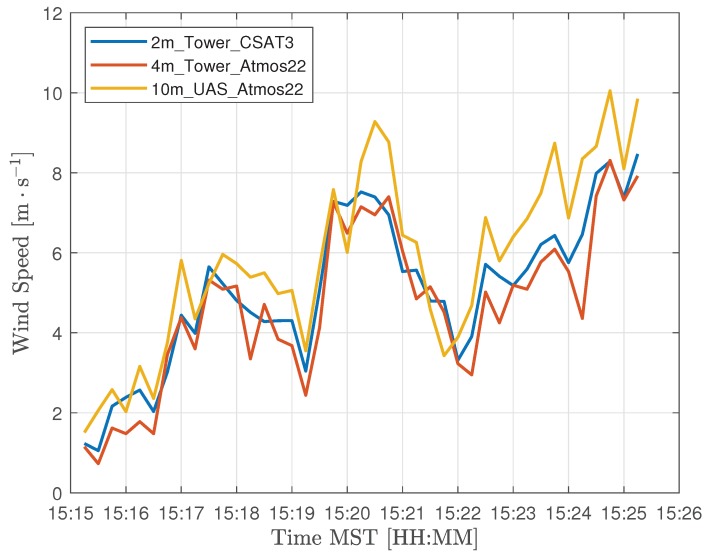
Comparison of wind speed measurements on the flux tower for a height of 2 m (CSAT3 sonic anemometer, yellow), 4 m (Atmos22 sonic anemometer), and 10 m (UAS with Atmos 22 sonic anemometer.

**Figure 8 sensors-18-04448-f008:**
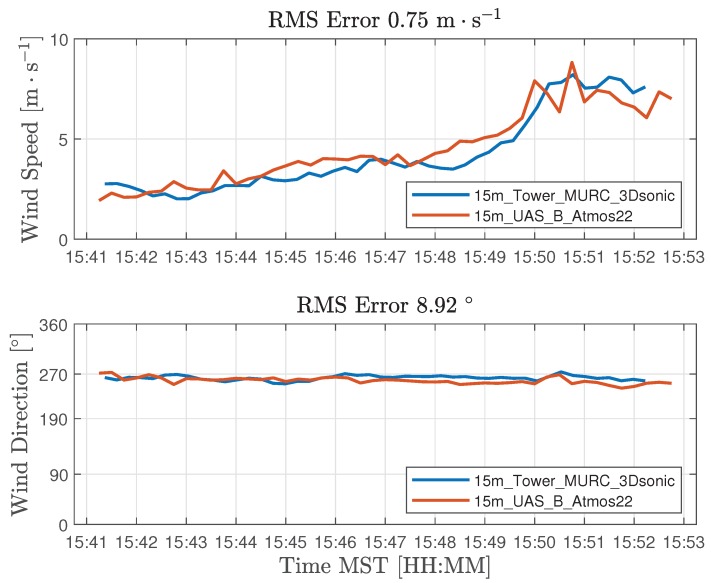
Root-mean-square (RMS) error comparison for wind speed and direction measurements collected from UAS B and MURC Tower at 15 meters above ground level.

**Figure 9 sensors-18-04448-f009:**
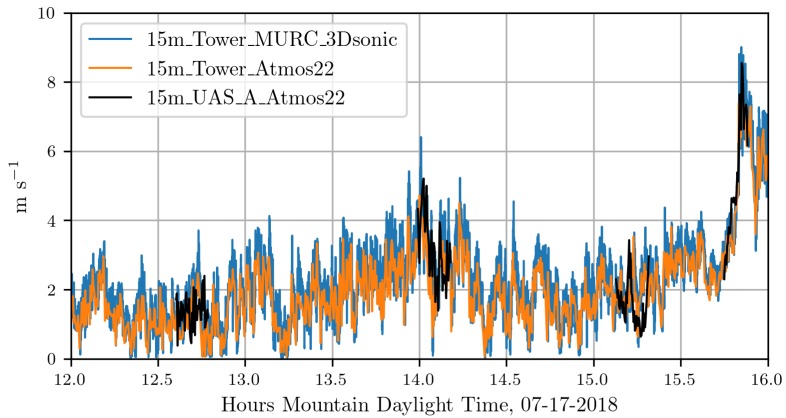
Comparison of wind speed measurements from UAS A and ground-based sensors.

**Figure 10 sensors-18-04448-f010:**
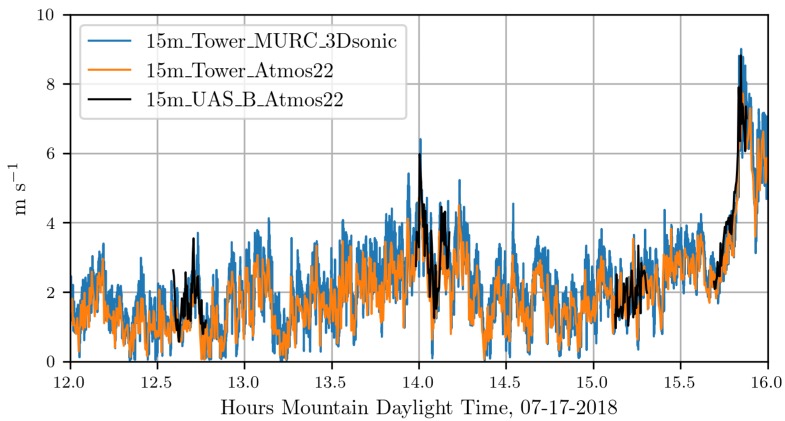
Comparison of wind speed measurements from UAS B and ground-based sensors.

**Figure 11 sensors-18-04448-f011:**
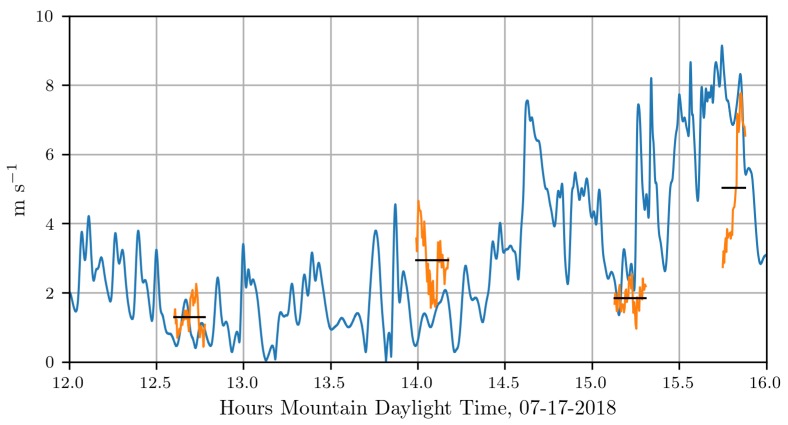
Wind speed from WRF-LES grid point nearest to where the attraction rate was calculated (blue) and wind speed as measured by sensors, interpolated to the attraction rate position (orange) overlaid with flight average of the wind speed (black). Wind speed from the WRF-LES comes from the 10 m height level, while wind speed measurements were performed by sensors at 15 m.

**Figure 12 sensors-18-04448-f012:**
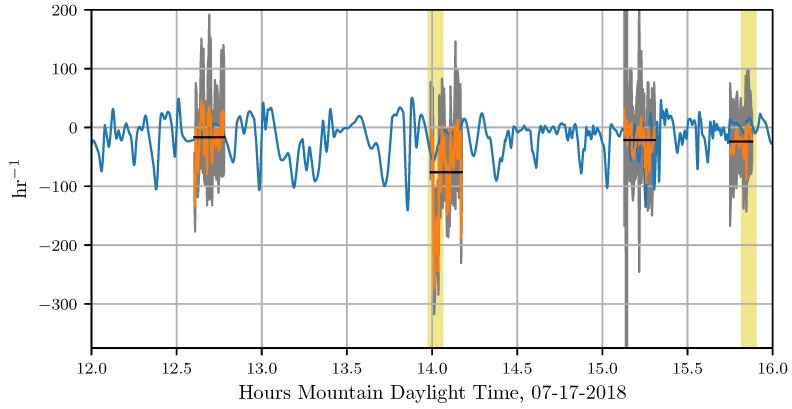
The attraction rate from WRF-LES predictions (blue) and the attraction rate as calculated from sensor measurements (orange) overlaid with flight average of the attraction rate (black). The uncertainty ranges for the attraction rate measurements are shown in gray. The attraction rate from the WRF-LES comes from the 10 m height level wind speed, while wind speed measurements used to calculate the attraction rate were performed by sensors at 15 m. Times of interest are highlighted with a yellow vertical line, corresponding to the predicted attraction rate fields shown in Figure 13 and Figure 14, respectively.

**Figure 13 sensors-18-04448-f013:**
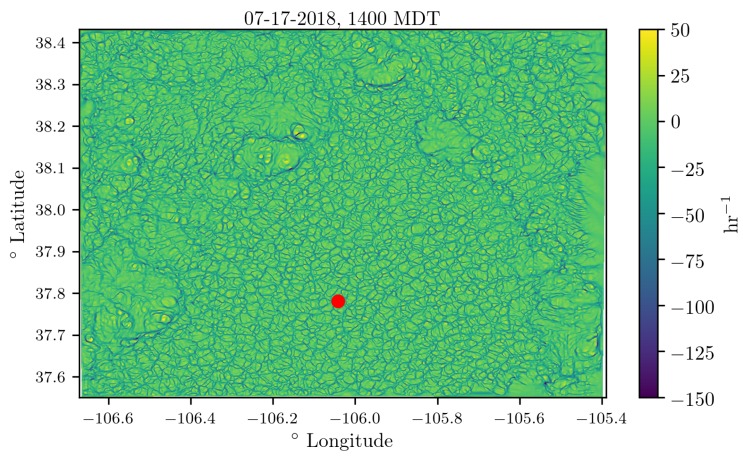
The attraction rate field at 1400 MDT, convective cells can be seen, bordered by troughs of the attracting field, throughout the domain. A front appears to be blowing an attracting feature out of the east of the domain. Sampling region is marked with a red dot.

**Figure 14 sensors-18-04448-f014:**
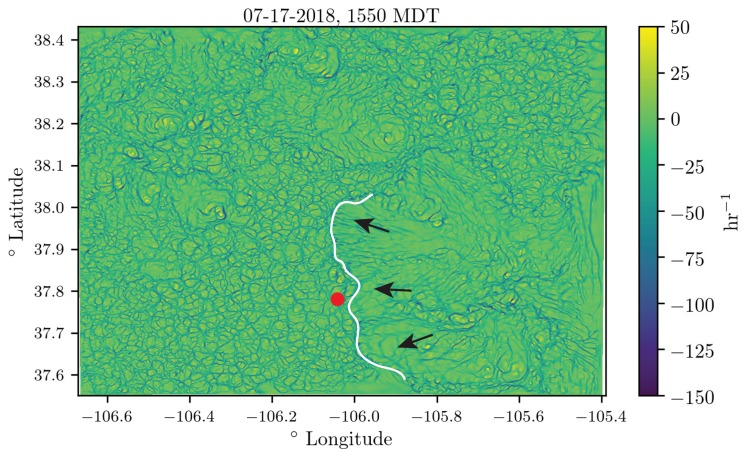
The attraction rate field at 1550 MDT, an attracting LCS can be seen along the center of the domain having ridden the front out of the east. The sampling region is marked with a red dot. Part of the attracting feature is shown as a white line. The direction the feature is moving is shown by black arrows.

**Figure 15 sensors-18-04448-f015:**
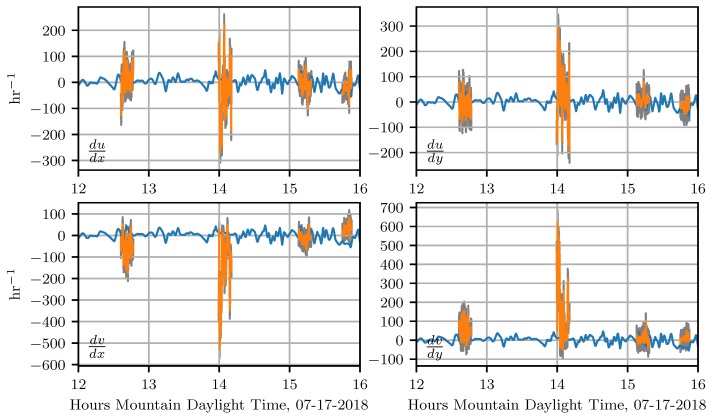
The velocity gradient from WRF-LES predictions (blue) and the velocity gradient as calculated from sensor measurements (orange). The velocity gradient from the WRF-LES comes from the 10 m height level wind speed, while wind speed measurements used to calculate the velocity gradient were performed by sensors at 15 m. The uncertainty ranges for the velocity gradient as calculated from sensor measurements are shown in gray.

**Table 1 sensors-18-04448-t001:** UAS mission and ground station details. UAS sensor packages were functionally identical. Time start and time end are in local time, Mountain Daylight Time. Height is in meters above ground level.

Sensor Package	Description of Operation	Date	Time Start	Time End	Height	Location	Lat	Long
UAS_A2	Calibration flight w/MURC.	14 July 2018	13:30:21	13:41:05	15	MURC Tower	37.781914	−106.041412
UAS_A4	Coordinated flight w/ B4	14 July 2018	16:43:07	16:54:55	10	East Runway	37.780315	−106.040772
UAS_A5	Coordinated flight w/ B5	14 July 2018	17:15:58	17:27:19	10	East Runway	37.780312	−106.040763
UAS_A16	Coordinated flight w/ B7	16 July 2018	14:42:07	14:53:04	15	East Runway	37.780308	−106.040753
UAS_A17	Coordinated flight w/ B8	16 July 2018	15:18:01	15:28:17	15	East Runway	37.780336	−106.040746
UAS_A22	Coordinated flight w/ B9	17 July 2018	12:36:00	12:47:24	15	East Runway	37.780287	−106.04076
UAS_A23	Coordinated flight w/ B10	17 July 2018	13:59:29	14:10:59	15	East Runway	37.780307	−106.040763
UAS_A25	Coordinated flight w/ B11	17 July 2018	15:07:39	15:19:02	15	East Runway	37.780398	−106.040762
UAS_A26	Coordinated flight w/ B12	17 July 2018	15:41:53	15:53:11	15	East Runway	37.780338	−106.040762
UAS_B1	Calibration flight w/ Flux Tower.	13 July 2018	15:15:07	15:25:27	10	Above UK WS	37.781644	−106.039170
UAS_B2	Calibration flight w/ MURC.	14 July 2018	14:15:23	14:26:07	15	MURC Tower	37.78188077	−106.0414296
UAS_B4	Coordinated flight w/ A4	14 July 2018	16:42:10	16:52:40	10	West Runway	37.78155488	−106.0422978
UAS_B5	Coordinated flight w/ A5	14 July 2018	17:15:27	17:26:28	10	West Runway	37.7815583	−106.0422984
UAS_B7	Coordinated flight w/ A16	16 July 2018	14:43:05	14:52:42	18	West Runway	37.78156695	−106.0422614
UAS_B8	Coordinated flight w/ A17	16 July 2018	15:16:17	15:27:48	9	West Runway	37.78158549	−106.0422597
UAS_B9	Coordinated flight w/ A22	17 July 2018	12:35:22	12:46:56	15	West Runway	37.78153018	−106.0422848
UAS_B10	Coordinated flight w/ A23	17 July 2018	13:58:58	14:10:31	15	West Runway	37.78155617	−106.0422905
UAS_B11	Coordinated flight w/ A25	17 July 2018	15:07:12	15:18:49	15	West Runway	37.78156052	−106.0422956
UAS_B12	Coordinated flight w/ A26	17 July 2018	15:41:12	15:52:55	15	West Runway	37.78156436	−106.0422941
Ground1		13 July 2018	11:45:00	15:55:00	4	On Flux Tower	37.781644	−106.03917
Ground2		14 July 2018	8:00:00	18:30:00	4	On Flux Tower	37.781644	−106.03917
Ground3		15 July 2018	11:00:00	14:45:00	4	On Flux Tower	37.781644	−106.03917
Ground4		16 July 2018	8:40:00	15:35:00	15	On MURC Tower	37.782097	−106.041412
Ground5		17 July 2018	8:15:00	16:00:00	15	On MURC Tower	37.782005	−106.041504

**Table 2 sensors-18-04448-t002:** Pearson correlation coefficients for wind speed measurements between different UAS packages and ground-based sensors.

	15m_Tower_MURC_3Dsonic	15m_Tower_Atmos22	15m_UAS_A_Atmos22	15m_UAS_B_Atmos22
**15m_Tower_MURC_3Dsonic**	–	0.970	0.876	0.914
**15m_Tower_Atmos22**		–	0.868	0.895
**15m_UAS_A_Atmos22**			–	0.868
**15m_UAS_B_Atmos22**				–

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
