# Peer review of "Coordinated Unmanned Aircraft System (UAS) and Ground-Based Weather Measurements to Predict Lagrangian Coherent Structures (LCSs)"

_sensors, 2018, doi:10.3390/s18124448_

Round 1
Reviewer 1 Report
The paper focuses the wind speed and direction estimation using two UAS and a ground measurement system. The experiment is described in a very detailed way and the authors report extensive measurements of the atmospheric properties deploying a considerable amount of human and hardware support. The measurements have been carefully examined and the objective to detect a Lagrangian Coherent Structure (LCS) was successfully attained. The paper is very well written an only a small misspell was detected (wind gradient trensor), above eq3. Overall the detailed information reported are very useful, but a few tables (e.g. Table 1) contain too much detail regarding the objectives of the paper. One important issue I found missing is a discussion about the position accuracy of the UAS system and how it impacts the estimation of the wind gradient tensor.
Author Response
Reviewer Comment: The paper is very well written an only a small misspell was detected (wind gradient trensor), above eq3.
Author Response: We have fixed the typo.
Reviewer Comment: Overall the detailed information reported are very useful, but a few tables (e.g. Table 1) contain too much detail regarding the objectives of the paper.
Author Response: We have reduced Table 1 to focus only on the flights reported in the paper.
Reviewer Comment: One important issue I found missing is a discussion about the position accuracy of the UAS system and how it impacts the estimation of the wind gradient tensor.
Author Response: We have added uncertainty measurements in Figures 12 and 13, and have addressed the spatial accuracy in the discussion.
Reviewer 2 Report
The description not relating with this study should be omitted. In Abstract, "Thirty-eight flights were conducted 10 to 80 meters above ground level (AGL)". This study used just 15 m AGL flight data. Table 1 contains a lot flight data not used in this study.
Temperature measurements are not related with the objective of this study. Please justify the inclusion of description of temperature measurements into this paper.
I don't think "the WRF-LES model provided fairly accurate predictions" (Line 304). I can see the large differences between WRF wind (Fig 13) and tower (Fig11 12). Please discuss WRF-LES simulation validity deeper.
There is no discussion on the accuracy of spatial wind speed gradient estimation by this experiment setting
Author Response
Reviewer Comment: The description not relating with this study should be omitted. In Abstract, "Thirty-eight flights were conducted 10 to 80 meters above ground level (AGL)". This study used just 15 m AGL flight data. Table 1 contains a lot flight data not used in this study.
Author Response: We have reduced Table 1 and the text of the paper to focus only on the flights reported in the paper.
Reviewer Comment: Temperature
measurements are not related with the objective of this study. Please
justify the inclusion of description of temperature measurements into
this paper.
Author Response: We have removed one table and two figures related to the temperature measurements.
Reviewer Comment: I don't think "the WRF-LES model provided fairly accurate predictions" (Line 304). I can see the large differences between WRF wind (Fig 13) and tower (Fig11 12). Please discuss WRF-LES simulation validity deeper.
Author Response: We have provided additional discussion of the WRF-LES model and have addressed uncertainties.
Reviewer Comment: There is no discussion on the accuracy of spatial wind speed gradient estimation by this experiment setting.
Author Response: We have added uncertainty measurements in Figures 12 and 13, and have addressed the spatial accuracy in the discussion.